# Effects of Functionalized Iron Oxide Magnetic Nanoparticle Suspensions on Seed Morphology and Physiology in Yellow Maize and Chili Pepper

**DOI:** 10.3390/plants14233592

**Published:** 2025-11-25

**Authors:** Álvaro Velásquez, Jeaneth Urquijo, Yessica Montoya, Danna Susunaga, Diego Villanueva

**Affiliations:** 1Grupo de Materiales y Electromagnetismo, Universidad EAFIT, Medellín 050022, Colombia; 2Grupo de Estado Sólido, Universidad de Antioquia, Medellín 050010, Colombia; 3Grupo de Investigación en Biociencias y Tecnología, Universidad EAFIT, Medellín 050022, Colombiadvillanu@eafit.edu.co (D.V.)

**Keywords:** magnetite–maghemite, nanocomposites, maize, chili pepper, phytotoxicity, morpho-anatomical and physiological parameters

## Abstract

We evaluated the effects of suspensions of magnetite–maghemite nanocomposites functionalized with quaternized chitosan and phosphate groups on morpho-anatomical and physiological traits of yellow maize (*Zea mays*) and chili pepper (*Capsicum annuum*) seeds. A phytotoxicity assay was first conducted by applying aqueous suspensions of these nanocomposites to maize seeds at iron concentrations up to 100 ppm, using deionized water as the control under humid chamber conditions. After incubation, seeds treated with concentrations above 100 ppm exhibited reductions in biomass and root length compared with the control, suggesting phytotoxicity at high levels. Based on these results, suspensions containing 25, 35, and 45 ppm of iron, with corresponding phosphorus concentrations of 2.9, 4.0, and 5.2 ppm, were selected for a second in vitro assay using both maize and chili seeds. No statistically significant differences between treatments and control were detected for the variables measured. Germinated seeds from this assay were further evaluated under greenhouse conditions, where measured parameters also showed no significant differences between treatments and control in either crop. Overall, the findings indicate that aqueous suspensions of magnetite–maghemite nanocomposites with iron concentrations below 100 ppm do not produce phytotoxic effects on seed germination or morpho-anatomical and physiological traits measured.

## 1. Introduction

The use of nanoparticles (NPs) has expanded rapidly across diverse fields [1,2]. In agriculture in particular, NPs hold the potential to transform food production techniques, as they can serve as carriers of bioactive agents (such as growth factors, pesticides, and fungicides) directly to plants [3]. Within the classification of NPs is the category of metal NPs, which are synthesized from essential metals and are among the most widely manufactured types of nanomaterials, owing to their unique physical and chemical properties [4,5,6]. Of special interest in plant protection and fertilization experiments are the iron oxide nanoparticles of magnetite (Fe_3_O_4_), maghemite (γ-Fe_2_O_3_), a highly oxidized state of magnetite, and hematite (α-Fe_2_O_3_), which have exhibited contrasting effects, from beneficial to phytotoxic, across various crops. Some studies have reported that iron oxide NPs can increase seed germination index in tomato [3] and wheat [7] seeds, seedling vigor, plant biomass, and yield and enhance physiological function [3,8], showing that these NPs can have a similar or even greater effect than commonly used fertilizers [9,10].

In contrast, other studies have reported decreases in shoot and root length and chlorophyll production in tobacco plants treated with 5 nm magnetite nanoparticles [11]. Studies on plants of *Arabidopsis thaliana* treated with Fe_2_O_3_ nanoparticles with either positive or negative surface charges [12] showed that at 25 mg/L, positively charged nanoparticles reduced seedling length by 20% and root length by 48%, whereas negatively charged nanoparticles produced less pronounced effects. In addition, a reduction of up to 11% in seed yield and lower pollen viability were observed. Studies on the foliar application of nanohematite to wheat seedlings (60 µg and 180 µg per plant) over 21 days [13] showed these nanoparticles induced excessive production of •OH radicals (Fenton-type), chlorophyll degradation, and disruption of photosynthetic mechanisms, which ultimately inhibited biomass production. Studies on the exposure of aquatic plants *Lemna minor* to iron oxide nanoparticles of akageneite (β-FeOOH) and hematite [14] showed these nanoparticles caused complete leaf death within 7 days, a dose-dependent increase in malondialdehyde peroxide and a reduction in chlorophyll content.

Different studies in this field agree that the effects of nanoparticles on living organisms are not yet fully understood [15], and further efforts are required to obtain evidence that will help determine the most suitable types and concentrations of nanomaterials for specific plant species. There is great expectation for the development of nanomaterials with potential for controlled nutrient release in plants, increasing their morphological and physiological parameters, as well as their resistance against pests, plagues, and lack of nutrients in the soil, among other adverse effects [16,17]. A promising approach to achieve biocompatibility, biodegradability, and colloidal stability, as well as to modulate potential toxic or agglomerative effects of NPs in physiological environments, is their surface functionalization with biocompatible materials to obtain core–shell nanocomposites, whose chemical structure can be tailored to improve solubility at physiological pH values [18,19]. An example of this is chitosan, an amino polysaccharide obtained from the deacetylation of chitin, a polysaccharide found mainly in the exoskeletons of crustaceans. In addition to being biocompatible and biodegradable, chitosan possesses antifungal and antimicrobial activity, which highlights its potential in biotechnology. However, for use in biological systems, its structure must be chemically modified to overcome its low solubility at physiological pH.

To our knowledge, few studies have investigated the impact of chitosan-coated iron oxide nanoparticles on maize (*Zea mays*) and chili pepper (*Capsicum annuum*), two crops of great importance for global food security and human health. Most published nanotoxicology research has instead focused on mammalian cytotoxicity or on effects in animals and bacteria. The purpose of this study is to address two fundamental questions: 1. Are magnetite–maghemite nanoparticles functionalized with quaternized chitosan biocompatible with yellow hybrid maize and chili pepper seeds within a given iron concentration range? 2. Does the functionalization of magnetite–maghemite nanoparticles with quaternized chitosan and phosphates enhance the morphological and physiological parameters of seeds and their subsequent seedlings compared with the water control?

Magnetite and its oxidized form, maghemite, are preferred in this study due to their superior biocompatibility and stronger magnetic response compared to other iron oxides such as hematite (α-Fe_2_O_3_), goethite (α-FeOOH), and akaganeite (β-FeOOH) [20,21,22]. Even though the synthesis method employed in this study was intended to produce a single phase of magnetite, the natural oxidation of magnetite to maghemite is inevitable under atmospheric conditions after the synthesis process. As a result, a mixture of both phases is commonly obtained. Moreover, the magnetic behavior of magnetite and maghemite is particularly advantageous for biotechnology applications, including drug delivery, magnetic hyperthermia, and the targeted transport of nanomaterials within living systems. In contrast, hematite, goethite, and akaganeite exhibit antiferromagnetic ordering, resulting in a much weaker magnetic response that limits their suitability for biotechnological applications. Details of the experiments and results obtained are presented in the following sections of this paper.

## 2. Materials and Methods

### 2.1. Precursor Reagents for the Synthesis of the Nanocomposites

For the synthesis of the nanocomposites, the following Merck-brand (St. Louis, MO, USA) reagents were used: Chitosan (degree of deacetylation > 75%), Glycidyl trimethylammonium chloride (GTMAC), Perchloric acid HClO_4_ (70–72%, CAS 7601-90-3), FeCl_3_·6H_2_O, Fe(NH_4_)_2_(SO_4_)_2_·6H_2_O, Sodium hydroxide NaOH and oxalic acid C_2_H_2_O_4_. All reagents were analytical grade and used as received. Phosphate rock was obtained from a Colombian mine.

### 2.2. Preparation of N-2-Hydroxy-Propyl-3-Trimethyl Ammonium Chitosan Chloride (HTCC)

Quaternization of chitosan is a process necessary to increase the solubility of this polymer in the alkaline conditions required to obtain magnetite–maghemite for coprecipitation. HTCC was synthesized by a reaction with GTMAC at 80 °C, employing the method reported by Ruihua et al. [23]. For this purpose, 1.0 g of chitosan was suspended in 150.0 mL of deionized water at 60 °C and dissolved by the addition of 1.0 mL of HClO_4_ under magnetic stirring at 500 rpm. Afterward, 8.0 mL of GTMAC was added in two portions at intervals of 0.5 h. Subsequently, the reaction continued at 80 °C for 8 h. The HTCC obtained was precipitated from the reaction solution with acetone, and it was dried and resuspended in deionized water for its subsequent use.

### 2.3. Preparation of Aqueous Suspensions of Magnetic Nanocomposites Coated with Quaternized Chitosan

Magnetite–maghemite nanoparticles functionalized with quaternized chitosan (Fe_3_O_4_-QC) were prepared by chemical coprecipitation of Fe^3+^ and Fe^2+^ species in the presence of a quaternized chitosan solution. As a first step, 0.500 g of quaternized chitosan was dissolved in 100.0 mL of deionized water, and the solution was bubbled with N_2_ (g) grade 5.0 for 30 min. Afterward, the pH of the solution was adjusted and maintained at 10 with the controlled addition of a solution 1.0 M in NaOH, using a SI Analytics TitroLine 7000 titrator. Subsequently, a solution containing 0.500 g of Fe(NH_4_)_2_(SO4)_2_·6H_2_O and 0.700 g of FeCl_3_·6H_2_O were added, and a black suspension was obtained. The suspension obtained was left under magnetic stirring for one hour and subsequently washed by dialysis until a pH = 6.5 was reached.

### 2.4. Binding of PO_4_^3−^ Ions to the Surface of Fe_3_O_4_-QC

For obtaining the PO_4_^3−^ ions to be bonded to the Fe_3_O_4_-QC surface, 2.000 g of phosphoric rock was dissolved in 100.0 mL of a 2% solution in oxalic acid. The mixture was heated for 4 h at 80 °C, then allowed to stand for 24 h and filtered. Afterward, 50.0 mL of this solution was mixed with 100.0 mL of the Fe_3_O_4_-QC suspension and stirred for 1 h. The suspension obtained was washed by dialysis, and the concentrations of phosphates and iron in the solution were determined by UV–Vis spectrophotometry measurements. According to UV–Vis measurements, the maximum phosphorus content adsorbed on the surface of nanoparticles was 40 ppm. Figure 1 shows the suspension obtained, as well as the subsequent washing process by dialysis. No precipitation was observed during the subsequent stages of the experiment. Zeta potential measurements presented in Section 3.1.3 show a monomodal distribution centered at 57 mV, indicating good colloidal stability of the suspension, supported by the positive surface charge of the nanocomposites.

### 2.5. Characterization of Magnetic Nanocomposites

The synthesized nanocomposites were characterized by structural, chemical and magnetic techniques. X-ray measurements were taken with a PANalytical X’Pert PRO diffractometer, with Cu-K_**α**_ radiation (*λ* = 1.540598 Å), angular range 2θ: 15–80°, step of 0.05° and counting time of 1 s per step. Fourier transform infrared (FTIR) spectroscopy measurements were obtained with a PerkinElmer Spectrum Two infrared spectrometer. Room temperature transmission Mössbauer measurements were taken with a Mössbauer spectrometer developed at the Laboratory of Instrumentation and Spectroscopy of the University EAFIT [24], which operates in the constant acceleration mode, with a radioactive source of 57Co/Rh with initial activity of 25 mCi and speeds between −12 and 12 mm s^−1^. Mössbauer absorbers were prepared by diluting the sample in sugar until an effective thickness of 6 mg-Fe cm^−2^ was obtained. Magnetization curves were taken with a vibrating sample magnetometer developed at the Instrumentation and Spectroscopy Laboratory of EAFIT University, with a resolution of 3 × 10^−7^ A m^2^ in magnetic moment and a range of 438 kA m^−1^ in magnetic field. The iron and phosphorus contents in the aqueous suspensions of MNPs were quantified by UV–Vis spectroscopy, with a spectrophotometer Thermo Scientific GENESYS 150 UV–Vis.

### 2.6. Preparation of the Aqueous Suspensions of Nanocomposite

To adjust the concentrations of iron and iron–phosphorus supplied by the nanocomposite suspension for the in vitro and greenhouse assays, the initial suspension was diluted with deionized water. In the phosphate-functionalized nanocomposite suspensions, the maximum phosphorus concentration was 5.2 ppm. The phosphorus concentration is limited by the capacity of the nanocomposites to adsorb phosphate groups onto their surface. Although the maximum phosphorus concentration loaded onto the diluted nanocomposite suspension is lower than the 20–40 ppm phosphorus levels recommended for fertilization processes [25], this study allows us to determine whether the high surface-to-volume ratio of the nanoparticles within the nanocomposite confers any advantage as a macronutrient carrier at phosphorus concentrations below those reference levels. Table 1 lists the labels assigned to the aqueous nanocomposite suspensions according to their iron and phosphorus concentrations.

### 2.7. Seeds Preparation

Seeds of yellow hybrid maize, reference FNC31AC, which were supplied by “Federación Nacional de Cultivadores de Cereales, Leguminosas y Soya (FENALCE)” from Colombia, is a biofortified variety of maize containing the *opaco-2* gene, which allows it to synthesize a high concentration of lysine and tryptophan, essential amino acids for high protein quality, which means that these seeds have superior biological characteristics to normal maize seeds [26]. Chili pepper seeds were obtained from an agricultural supermarket (Tierragro) located in Medellín, Colombia. They belong to a species called *Capsicum annuum* L., which grows in tropical climates and possesses a range of bioactive compounds and essential nutrients that give this crop an important position in the economic world [27]. Seeds were disinfected before being used in the different assays. Initially, seeds were rinsed with a solution of iodopovidone, soap and water. Afterward, the maize seeds were immersed in 15% sodium hypochlorite for 3 min, and the chili pepper seeds were immersed in 2% sodium hypochlorite for 15 min. Then, seeds were rinsed three times with sterile deionized water. Finally, the seeds were left to dry in a laminar flow chamber for approximately 10 min.

### 2.8. In Vitro Phytotoxicity Assays

The first in vitro experimental tests were performed by incubation of seeds in a humid chamber, which consisted of placing 10 maize seeds per Petri dish (100 mm × 15 mm) with a cellulose filter paper at the bottom impregnated with 3 mL of deionized water (control) or with a suspension of nanocomposite with different iron concentration, namely C12.5-0, C25-0, C50-0, C75-0, and C100-0. Petri dishes were stored in a room with the lowest light incidence possible at 23 °C for 8 days. After this time, the biomass (g) and root length (cm) of germinated seeds were recorded. Based on the results, a second in vitro test was carried out under the same experimental conditions as above, evaluating the effect of concentrations of nanocomposites C25-3, C35-4, and C45-5 on the germination percentage (%) and root length (cm) of maize seeds (10 seeds per Petri dish) and chili pepper seeds (20 seeds per Petri dish). Each treatment was carried out in triplicate.

### 2.9. Greenhouse Assay

Germinated seeds in the in vitro assay, treated with suspensions C25-3, C35-4, and C45-5, were sown in pots and seedling trays with a black commercial soil (mixture of organic soil, decomposed plant matter, and essential minerals; obtained from Tierragro) for maize and chili pepper, respectively. For maize, we prepared three pots per treatment, with three germinated seeds in each one, while in the seedling trays, we prepared three chili pepper seeds per cell, with four cells per treatment. Both crops were incubated in a greenhouse at an average temperature of 28 °C for 28 days, watered once a day at 9 a.m. with 250 mL of tap water per maize pot and 100 mL per chili pepper seedling tray, and at the end of the day, the soil was checked to ensure it was not dry. After this time, the maize and chili pepper seedlings were harvested, and the following measurements were performed: dry biomass (mg), root length (cm), root thickness (mm), stem thickness (mm), seedling height (cm), and length of the largest leaf (cm).

### 2.10. Total Oxidizable Organic Carbon Measurements

The maize and chili pepper seedlings obtained in the greenhouse trial were sent to the laboratory of Grupo Interdisciplinario de Estudios Moleculares (GIEM, ICA Registration No. LB0000152021) of the Universidad de Antioquia, in Medellín-Colombia, which provided the service for the quantification of the Oxidizable Organic Carbon of these samples. The method used for this measurement was the Walkley–Black method, which is based on the conversion of organic carbon to inorganic carbon as carbon dioxide (CO_2_) [28].

### 2.11. Statistical Analysis

The data of the measurements in the first in vitro assay were evaluated by regression analysis using the Statgraphics software, version 19, in which the parameter values were considered to determine the relationship between the dependent variables (biomass, root length) with respect to the independent variable (iron content supplied by the suspension of nanocomposite). The data of second in vitro assay (three replicates), and the greenhouse trial (*n* = 3 for maize and *n* = 4 for chili pepper, per treatment), were analyzed using one-way analysis of variance (ANOVA), if normality, homoscedasticity and independence assumptions were satisfied, to determine significant differences between treatments, followed by a Dunnet’s and Tukey’s test for multiple comparisons at a 95% confidence level. Statistical analysis was performed with R Software version 4.2.2. The scripts used for data analysis in R can be found in the Appendix A.

## 3. Results and Discussion

### 3.1. Characterization of the Nanocomposite

#### 3.1.1. X-Ray Measurements

Figure 2 shows the X-ray diffraction pattern of the synthesized nanocomposite. Rietveld refinement was performed using the MAUD software 2.8 (Material Analysis Using Diffraction, version 2.8) [29] to determine structural parameters and estimate the relative phase composition of the sample, as summarized in Table 2.

The diffractogram reveals only inverse spinel phases corresponding to magnetite and maghemite, which are corroborated by the sequence of indexed peaks.

X-ray diffraction measurements are consistent with Mössbauer spectroscopy measurements presented in Appendix A, which show that the crystalline phase of the nanocomposite synthesized is composed of a mixture of magnetite (Fe_3_O_4_) and its highly oxidized state of maghemite (γ-Fe_2_O_3_) in equal percentages, being the maghemite phase in a superparamagnetic state due to small particle size effects. The presence of maghemite is expected due to the natural oxidation of Fe^2+^ to Fe^3+^ under atmospheric conditions following the synthesis process, having magnetite and maghemite similar biocompatibility [30].

#### 3.1.2. FTIR Measurements

Figure 3 presents the infrared spectrum of the nanocomposite; this spectrum exhibits several bands characteristic of Fe-O bonds at tetrahedral and octahedral sites of magnetite–maghemite, as well as polysaccharide chains of chitosan, among them Ether (C-O-C), Amide (-NH-CO) and C-H. This spectrum also shows a band located around 1478 cm^−1^, characteristic of the methyl group (-CH_3_). The presence of the methyl group evidences the quaternization of the chitosan bonded to the surface of the spinel nanoparticles.

#### 3.1.3. Zeta Potential and Dynamic Light Scattering Measurements

Figure 4a shows the zeta potential measurements of an aqueous suspension of the nanocomposite, performed at pH = 6.5. A single peak centered at +57 ± 2 mV is observed, based on three independent measurements per sample. This peak indicates a sufficiently positive surface charge on the nanocomposite, conferred by the quaternary amino groups of the quaternized chitosan, which ensures good colloidal stability and prevents particle aggregation [31]. The particle size distribution as a function of hydrodynamic diameter, shown in Figure 4b, exhibits a single, well-resolved peak centered at approximately 38 nm. This feature is attributed to magnetite–maghemite nanoparticles coated with quaternized chitosan chains and surrounded by a solvation shell of water molecules.

#### 3.1.4. TEM Measurements

Figure 5 presents a TEM micrograph of the nanocomposite obtained and its particle size distribution, respectively. As can be seen from the micrograph, the particles exhibit low aggregation and a nearly spherical morphology, with a mean particle size around 10 nm, estimated by the lognormal fitting of the particle size distribution.

#### 3.1.5. Magnetization Measurements

The room temperature magnetization curve of the nanocomposite, presented in Figure 6, is characteristic of soft magnetic materials. The sample saturated at a magnetic field of 200 kAm^−1^ approximately.

The hysteresis parameters, presented in Table 3, show a low saturation magnetization of 8.48 kAm^−1^, this value being at least 11 times smaller than that of a crystalline and stoichiometric magnetite (92 A m^2^ kg^−1^), which can be attributed to the presence of the diamagnetic component of chitosan, which, according to thermogravimetric measurements represents 70% of the mass of the sample, as well as possible surface anisotropy effects related to the small particle size of the nanoparticles.

### 3.2. Application of the Nanocomposites in Maize and Chili Pepper Seeds

#### 3.2.1. In Vitro Phytotoxicity Assay

In this study, the germination percentage was used to evaluate potential phytotoxic effects at the morpho-anatomical level in plants, as this is a parameter commonly used for this purpose. Therefore, this variable was evaluated as shown in Figure 7 for the first in vitro assay, in which the application of suspensions of C12.5-0, C25-0, C50-0, C75-0, C100-0, and deionized water (control) to FNC31AC maize seeds was analyzed. The results revealed germination rates above 93% in all treatments, indicating good seed viability and vigor, regardless of iron concentration. This is supported by the *p*-value obtained (<0.05), which indicates that there is no statistically significant relationship between the germination percentage and iron concentration. In addition, the 95% confidence intervals for the treatment means overlap, suggesting no statistically significant difference between the treatments and indicating that neither is better than the other.

For biomass and root length of germinated seeds (Figure 8), two other important variables for the evaluation of phytotoxicity, a regression analysis showed that both variables have a statistically significant relationship with the iron concentration present in the suspensions, with a *p*-value less than 0.05 in both cases; however, the 95% confidence intervals of the treatment means in both variables indicate that there is no statistical difference between the treatments.

According to the parabolic regression model shown in Figure 7 and Figure 8, and based on similar results such as those reported by evening primrose (*Oenthera biennis* L.) which showed that by increasing the concentration of iron in α-Fe_2_O_3_ suspensions, a significant decrease in growth and germination indices occurred [32], and this other study [33], where increases were observed in the number of leaves, leaf area, stem thickness, dry biomass, and chlorophyll content of Moringa oleifera at iron concentrations from 0 to 29 ppm supplied by Fe_3_O_4_ nanoparticles, and a decrease of these variables at an iron concentration of 43 ppm in the Fe_3_O_4_ suspension, for the second in vitro assay of this study, by using the same methodology, three suspensions with iron concentrations less than C50-0 were chosen. namely C25-3, C35-4, and C45-5. In this assay, both maize and chili pepper seeds were evaluated (Figure 9). Considering that phosphorus is one of the most important nutrients involved in development and growth of plants [34], and that it may have low bioavailability in the soil [35], in this second in vitro assay, the phytotoxic effect that could be produced in seeds when treated with iron oxide nanoparticles with phosphates attached to their surface at concentrations no higher than 5.2 ppm was analyzed, with the aim of evaluating a more environmentally friendly approach to enhance the availability of this macronutrient for plants.

As a result of this assay, neither the germination percentage (Figure 10a,b) nor the root length (Figure 10c,d) showed statistically significant differences between the treated seeds and the control, as determined by ANOVA at a 95% confidence level. However, only maize treatments C35-4 and C45-5 showed statistically significant differences between them in the germination percentage.

According to Figure 10a, the germination percentage of the maize seeds showed a value of over 70%, with the C45-5 treatment having the lowest value (70%) and the C35-4 treatment the highest (93%), while the chili pepper seeds had a value of over 77%, with the C45-5 treatment having the highest value (92%) and the C25-3 treatment the lowest value (77%). Although the results of germination percentage of both crops suggest that there is no presence of phytotoxic effects on the treated seeds at the morpho-anatomical level, as no statistically significant differences are observed between the treatments with respect to the control, and in addition to this, they reached a value of over 50% germination, which classifies them as seeds with good viability [36], a reduction is evident in the germination percentage observed in maize compared to the first assay. Considering that light, temperature, and the internal conditions of each seed are relevant factors in the germination process [37,38], one possible reason for the difference between the two trials was that the two germination trials were conducted with different batches of maize seeds, supplied sequentially by the supplier as our experimental work progressed. 

Although both batches met the viability threshold and were handled under the same storage and experimental conditions, variation between seed batches may have led to physiological heterogeneity during the germination tests. Regarding root structure, no necrosis was observed in either crop. Finally, in both variables, there were no significant differences between the treatments and the control, so it can be concluded that the suspensions of the nanocomposite evaluated do not have a phytotoxic effect on maize and chili pepper seeds at the morpho-anatomical level. Positive or negative effects of the phosphate-functionalized nanocomposites on both germination percentage and root length, although not statistically verified with the number of individuals used per treatment in the two evaluated plant species, cannot be ruled out. Greater statistical power or longer-term monitoring of the plants would be required to draw definitive conclusions in this regard.

#### 3.2.2. Greenhouse Assay

To monitor the growth of the seedlings coming from the seeds treated with the nanocomposite suspension and evaluate the effect of these on the physiological level, maize and chili pepper seeds germinated under the C25-3, C35-4, C45-5 and control treatments in the second in vitro assay were sown in commercial soil, where they grew under greenhouse conditions. During this period, the seedlings were watered daily, as mentioned in the methodology Section 2.9, and constantly monitored to ensure they did not suffer from any pests or diseases that could affect their growth. After 28 days of growth, the seedlings were removed from the pots and the following measurements were taken: seedling length, stem thickness, root length, root thickness, length of the longest leaf, and dry biomass (Figure 11).

According to Figure 11, none of the measurements performed on the seedlings showed a statistically significant difference between the evaluated treatments compared to the control, since the *p*-value obtained in each of them was higher than the value defined for this analysis (0.05). Only in the root thickness variable in the chili pepper seedlings, a significant difference was identified between the C25-3 and C35-4 treatments by Tukey’s test. In addition to this, Figure 12 shows that the seedlings of both chili pepper and maize developed well during the days of the assay, as there was no evidence of necrosis in any of their organs or growth retardation, whereas the growth of plants, being autotrophic organisms, is due to the ability of their organs, especially their leaves, to absorb carbon dioxide from the atmosphere and the ability of their roots to absorb minerals and water from the soil and convert this into their food [39,40]. It could be suggested, based on the results obtained at greenhouse level under the conditions evaluated in this assay, that the aqueous suspensions of nanocomposites used, although do not improve morphological parameters of the plants treated with respect to control, do not impede the correct functioning of the roots and organs of the seedlings. As a consequence, the growth and development of the seedlings was carried out properly, so it is considered that the suspensions of nanocomposites used do not produce phytotoxic effects at the early physiological level in the seedlings.

### 3.3. Total Oxidizable Organic Carbon Measurement (TOC)

Organic carbon is the essential and predominant element in the growth and development of plants, since together with other macronutrients such as nitrogen, it forms the basic components of plant tissues. Carbon, together with oxygen and hydrogen, makes up the organic fraction of plants, representing between 90 and 95% of their dry biomass [41,42]. In the case of this evaluation, it can be observed that the quantification of the organic carbon of the seedlings obtained from the greenhouse test (Table 4) shows a slight variation among treatments in both crops, for maize, the TOC percentage ranged from 35.3 to 37.5, with the highest value observed in treatment C45-5 (37.5 ± 0.1%) and the lowest in C35-4 (35.3 ± 0.1%), while for chili pepper seedlings, the TOC percentage ranged from 30.5 to 32.7, with the highest value in treatment C25-3 (32.7 ± 0.1%). This small difference between treatments could suggest that nanoparticle treatments do not alter the carbon composition of the plant tissues, which could be related to the results observed in the greenhouse section, where the variables evaluated, including dry biomass, stem thickness, root length and thickness, and leaf length, did not show statistically significant differences between the treatments and the control. Finally, as in the greenhouse results, the TOC percentages could indicate that the evaluated concentrations of the nanoparticle suspension would not have a positive effect on the seedlings at a physiological and morphological level under the experimental conditions of this study, but at the same time could indicate that they do not have a phytotoxic effect on them.

## 4. Conclusions

### 4.1. On the Biocompatibility of Nanocomposite Magnetite–Maghemite Quaternized Chitosan

Measurements of germination, root length, and dry biomass confirmed that the synthesized magnetite–maghemite nanocomposite functionalized with quaternized chitosan is biocompatible with yellow hybrid maize seeds at iron concentrations below 100 ppm. Nevertheless, measurements of dry biomass and root length of germinated seeds suggest that suspensions of the nanocomposite with iron concentrations not exceeding 50 ppm yield the best results for these two indicators.

### 4.2. On the Enhancement of Morphological and Physiological Parameters in Maize and Chili Pepper Seeds and Seedlings Derived from Seeds Treated with Nanocomposites of Varying Iron–Phosphorus Concentrations

The nanocomposite, subsequently functionalized with phosphate groups, reaching phosphorus concentrations of up to 5.2 ppm, was also biocompatible with yellow hybrid maize and chili pepper seeds; however, it did not produce significant differences in the morphological or physiological parameters of the resulting seedlings compared with the water control.

### 4.3. Future Works

Future assays should be conducted to confirm potential positive or negative effects of the phosphate-functionalized nanocomposites on the morphological and physiological indicators of maize and chili pepper plants. These assays should aim to increase the statistical power of the experiment, extend the monitoring period of plant growth, modify the synthesis method to enable adsorption of phosphorus concentrations higher than 5.2 ppm, or combine these approaches. Overall, we believe this study provides valuable insights for future research focused on the development of magnetic iron oxide-based nanomaterials that can support the advancement of the agricultural sector.

## Figures and Tables

**Figure 1 plants-14-03592-f001:**
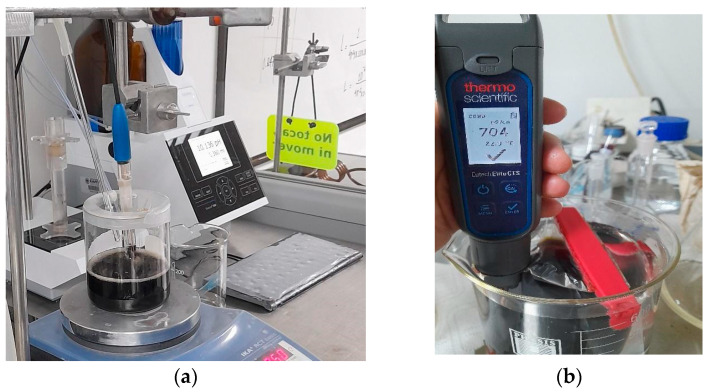
Synthesis of the nanocomposites: (**a**) View of the suspension of MNPs functionalized with chitosan and phosphates, (**b**) View of the washing of MNPs by dialysis.

**Figure 2 plants-14-03592-f002:**
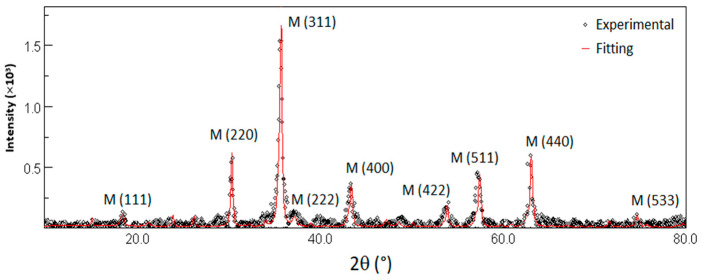
X-ray diffractogram of the synthesized nanocomposite.

**Figure 3 plants-14-03592-f003:**
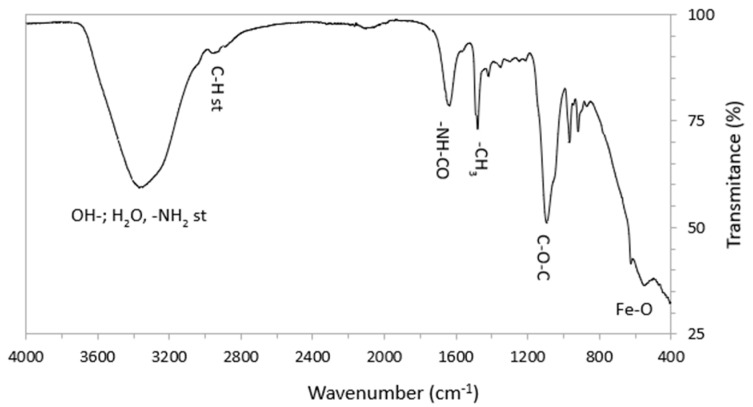
FTIR spectrum of the synthesized nanocomposite.

**Figure 4 plants-14-03592-f004:**
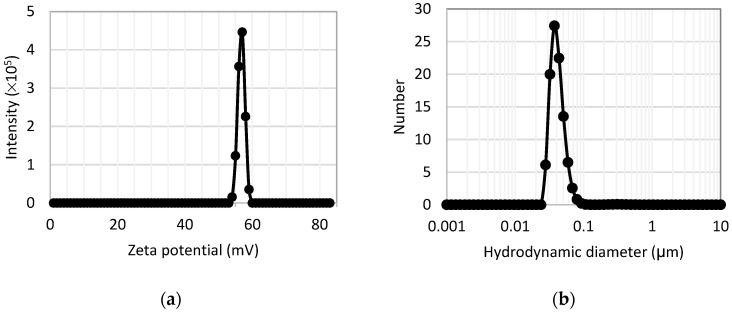
(**a**) Zeta potential measurement of the aqueous suspension of nanocomposite; (**b**) percentage number of particles per hydrodynamic diameter. Black dots represent experimental data, while solid curves represent interpolated experimental data.

**Figure 5 plants-14-03592-f005:**
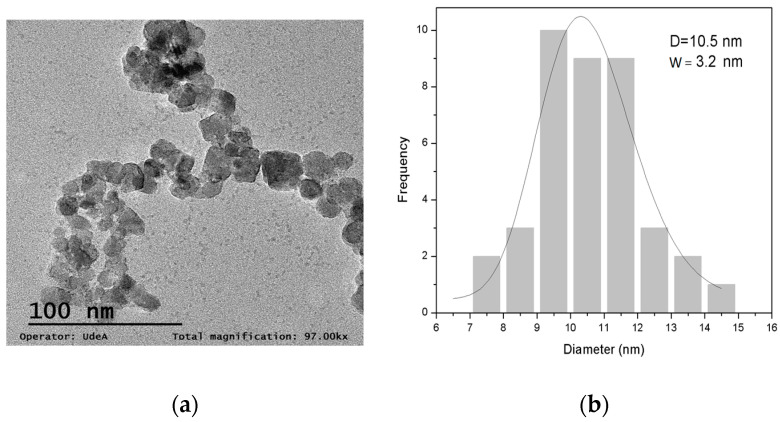
(**a**) TEM micrograph of the nanocomposite synthesized; (**b**) particle size distribution histogram.

**Figure 6 plants-14-03592-f006:**
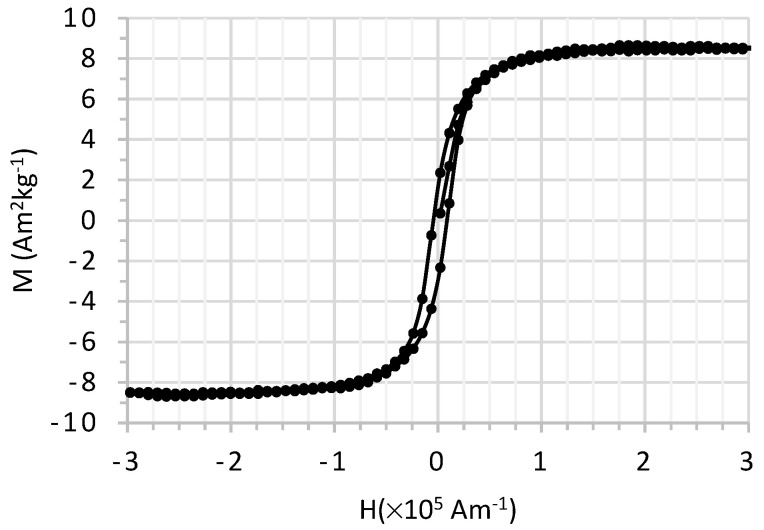
Magnetization curve of the nanocomposite.

**Figure 7 plants-14-03592-f007:**
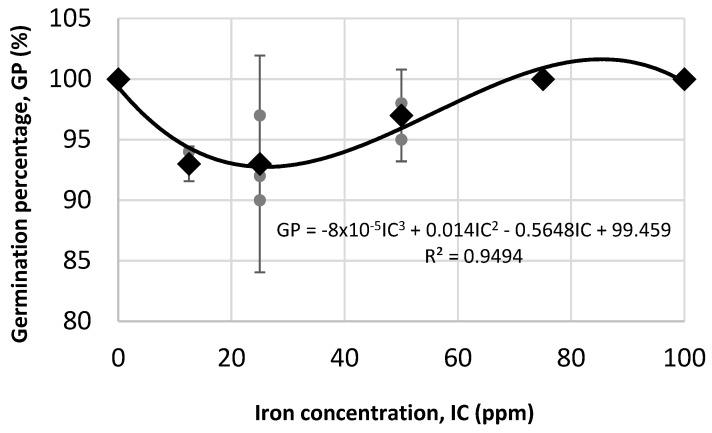
Regression analysis of different iron concentrations in the germination of maize seeds. Polynomial model grade 3, ANOVA of model: 0.066. Gray dots represent raw replicate points, black dots represent treatment means, and bars indicate margin of error with 95% confidence intervals calculated with the Student’s *t*-test.

**Figure 8 plants-14-03592-f008:**
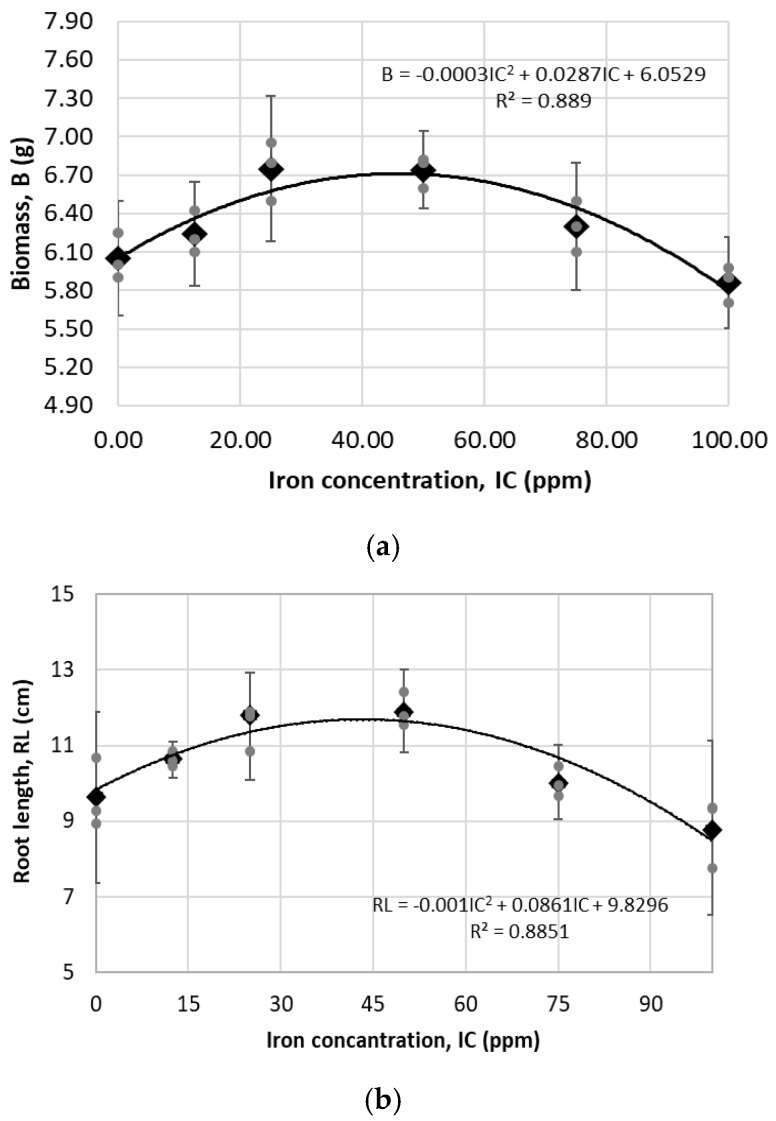
Regression analysis of different iron concentrations in germination of maize seeds: (**a**) Evaluation of variable Biomass (polynomial model grade 2, ANOVA of model: 0.037), (**b**) Evaluation of variable root length (polynomial model grade 2, ANOVA of model: 0.039). Gray dots represent raw replicate points, black dots represent the mean, and bars indicate the margin of error with 95% confidence intervals, calculated with Student’s *t*-test.

**Figure 9 plants-14-03592-f009:**
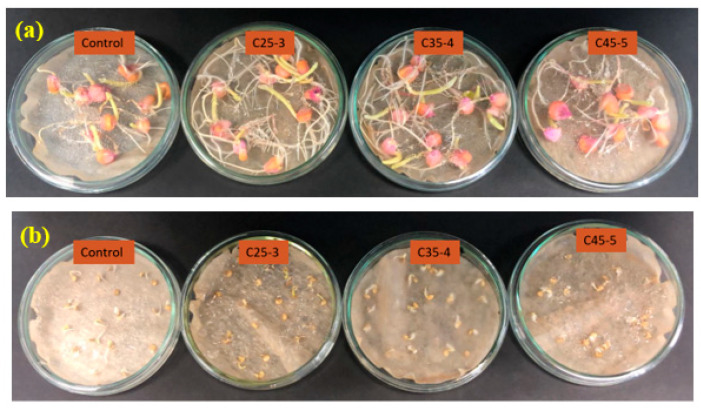
Evaluation of the effect of the iron content of nanocomposites at the in vitro level on (**a**) maize seeds and (**b**) chili pepper seeds. Images correspond to day 7 of the assay.

**Figure 10 plants-14-03592-f010:**
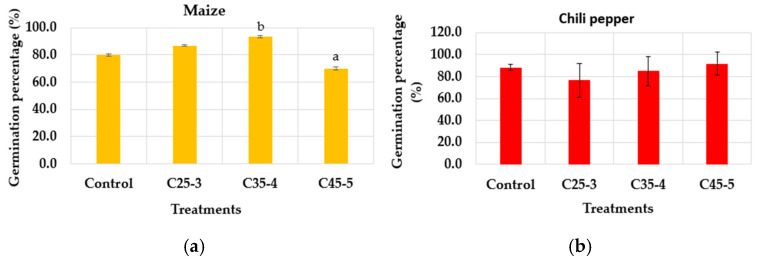
(**a**) Germination percentage of maize (*p*-value: 0.040); (**b**) germination percentage of chili pepper (*p*-value: 0.462); (**c**) root length of maize (*p*-value: 0.265), and (**d**) root length of chili pepper (*p*-value: 0.073) as a function of the iron concentration. ANOVA test with 95% confidence value. Different letters mean statistically significant differences between treatments (Tukey test, 95%).

**Figure 11 plants-14-03592-f011:**
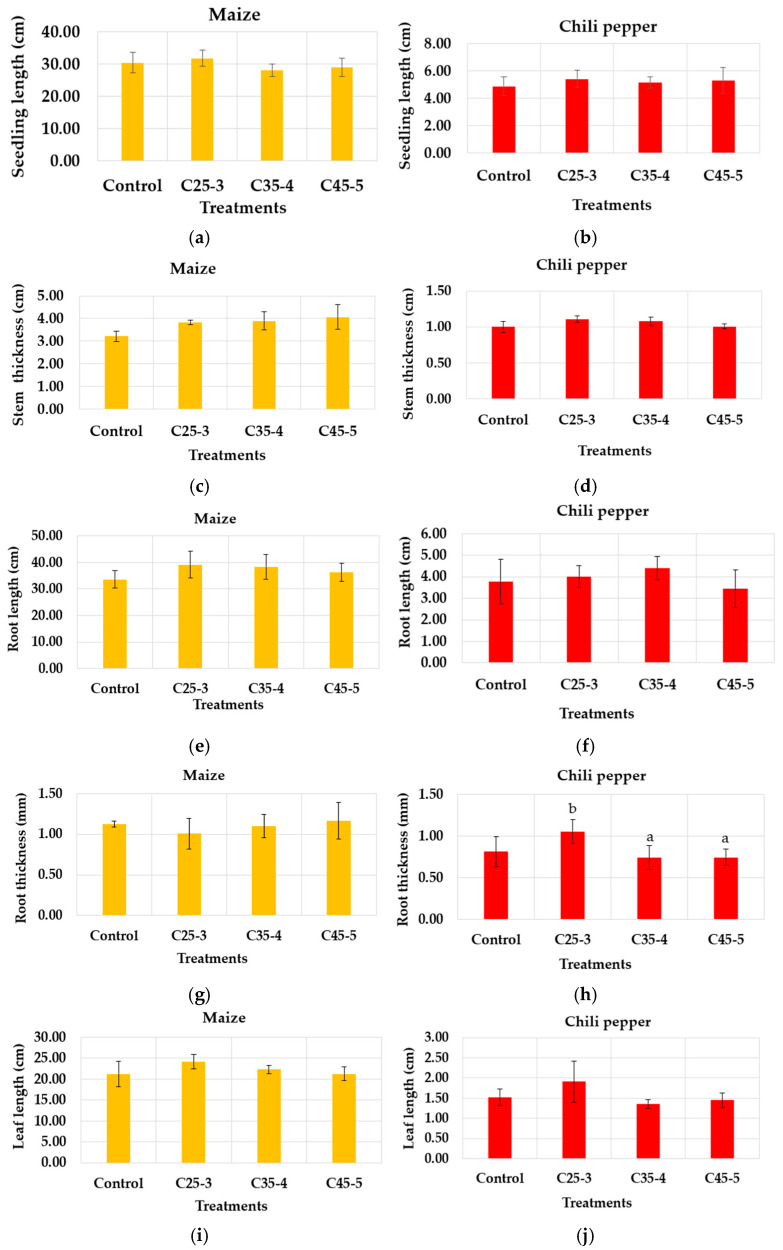
Graphs of morphological measurements carried out on maize and chili pepper seedlings grown under greenhouse conditions. (**a**) Seedling length for maize: *p*-value 0.392. (**b**) Seedling length for chili pepper: *p*-value 0.711. (**c**) Stem thickness of maize: *p*-value 0.081. (**d**) Stem thickness of chili pepper: *p*-value 0.075. (**e**) Root length of maize: *p*-value 0.417. (**f**) Root length of chili pepper: *p*-value 0.448. (**g**) Root thickness of maize: *p*-value 0.699. (**h**) Root thickness of chili pepper: *p*-value 0.041. (**i**) Leaf length of maize: *p*-value 0.311. (**j**) Leaf length of chili pepper: *p*-value 0.109. (**k**) Dry biomass of maize: *p*-value 0.063. (**l**) Dry biomass of chili pepper: *p*-value 0.782. ANOVA analysis, with a confidence value of 95%. Different letters mean statistically significant differences between treatments (Tukey test, 95%).

**Figure 12 plants-14-03592-f012:**
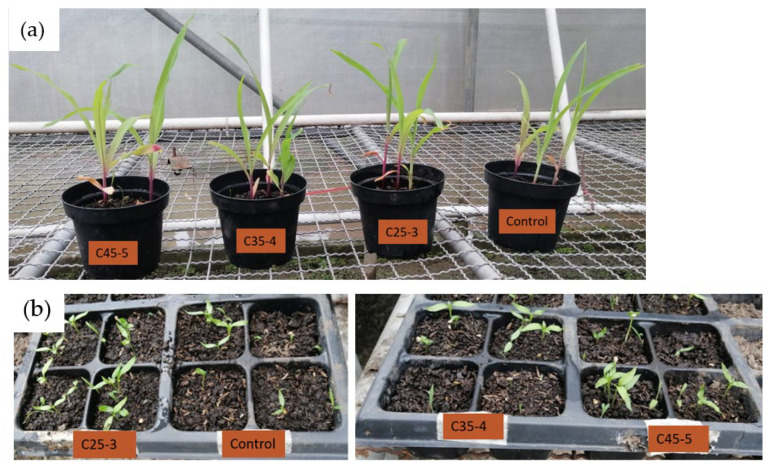
View of the seedlings derived from seeds treated with nanocomposites with different iron and phosphorus concentrations under greenhouse conditions. (**a**) Maize seedlings. (**b**) Chili pepper seedlings. Images correspond to day 15 of the assay.

**Table 1 plants-14-03592-t001:** Labels of the suspensions of nanocomposite used in the treatments according to their iron and phosphorus concentrations.

Suspension Label	Iron Concentration (ppm)	Phosphorus Concentration (ppm)
C12.5-0	12.5	0
C50-0	50	0
C75-0	75	0
C100-0	100	0
C25-3	25	2.9
C35-4	35	4
C45-5	45	5.2

**Table 2 plants-14-03592-t002:** Percentages and structural parameters of the spinel phase present in the nanocomposite, obtained by Rietveld refinement.

Phase	Percentage (%)	*a* (Å)	*D* (nm)
Magnetite	42 ± 13	8.377 ± 0.004	25 ± 5
Maghemite	58 ± 13	8.329 ± 0.002	18 ± 2

*a*—lattice parameter, *D*—mean crystallite diameter.

**Table 3 plants-14-03592-t003:** Hysteresis parameters of the synthetized nanocomposite.

Ms (A m^2^ kg^−1^)	Mr (A m^2^ kg^−1^)	Hc (kA m^−1^)
8.48 ± 0.01	1.90 ± 0.01	8.3 ± 0.2

Ms saturación magnetization, Mr remanent magnetization, Hc coercive magnetic field.

**Table 4 plants-14-03592-t004:** Measurements of total oxidizable organic carbon of maize and chili pepper seedlings under greenhouse conditions as a function of the iron and phosphorus concentrations supplied by the nanocomposite to the precursor seeds.

Treatment	Maize (%)	Chili Pepper (%)
Control	36.7 ± 0.1	30.8 ± 0.1
C25-3	35.6 ± 0.1	32.7 ± 0.1
C35-4	35.3 ± 0.1	31.3 ± 0.1
C45-5	37.5 ± 0.1	30.5 ± 0.1

## Data Availability

The original contributions presented in this study are included in the article/Appendix A. Further inquiries can be directed to the corresponding author.

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
