# Peer review of "Effects of Functionalized Iron Oxide Magnetic Nanoparticle Suspensions on Seed Morphology and Physiology in Yellow Maize and Chili Pepper"

_plants, 2025, doi:10.3390/plants14233592_

Round 1
Reviewer 1 Report
Comments and Suggestions for Authors
The work is devoted to the synthesis and study of phytotoxicity of magnetite-maghemite NPs functionalized with quaternized chitosan, and adsorbed on their surface phospahate. The topic fits within plant physiology, agronomy/agrochemistry, and the emerging field of agricultural nanomaterials.
Biological assays comprised two in vitro phytotoxicity tests. In vitro tests consisted of incubating maize and chili pepper seeds with bare nanocomposite suspension (12.5-100 ppm) or with adsorbed phosphate (2.9-5.2 ppm). They were followed by a greenhouse experiment where the germinated seedlings primed with nanocomposite suspension with adsorbed phosphate were grown in pots for 28 days and their biomass, root length/thickness, stem thickness, seedling height, and longest leaf length were measured. Lastly, total oxidizable organic carbon of the greenhouse-derived seedlings was quantified using the Walkley–Black method.
Across in vitro assays, maize and chilli pepper germination, biomass of the seedlings and root length did not change compared to the control up to 50 ppm of nanocomposite. In the subsequent greenhouse test, plant growth parameters also showed no treatment–control differences. These findings support the the overall conclusion that ≤45 ppm Fe and ≤5.2 ppm P were neither phytotoxic nor growth-promoting in these species. The obtained results are useful for early-stage risk assessment and dose selection in seed-priming/coating studies and for safety evaluation frameworks for agricultural nanomaterials.
Major comments:
- The authors should explicitly justify the selection of a magnetite–maghemite over alternative iron oxides (e.g., hematite, magnetite-only), outlining the hypothesized advantages.4
- At a maximum of 5.2 ppm P in the working suspensions (Table 1) and 3 mL per Petri dish, these are microgram-level doses; the authors should either justify that such P input is physiologically meaningful for priming/early growth or treat it as a limitation and consider increasing loading/concentration and quantifying release/uptake.
- The outcomes for maize and chili (Fig 4) should be replot as separate graphs using comparable y-scales within each species; the large between-species magnitude differences makes the current combined plot very hard to read.
- For each treatment in germination of maize seeds tests, raw replicate points and the mean with 95% confidence intervals should be overlayed ( 1), and state the CI method should be pointed in the caption. Based on that, the statistical evidence of the character of the dependence of dose–response should be found (is there an extremum still?)
- The authors’ conclusions about photosynthetic activity based on total oxidizable organic carbon (TOC) measured by the Walkley–Black method are not justified. Walkley–Black method is a soil wet-oxidation assay not validated for plant tissues as a proxy for photosynthesis, and TOC% is largely insensitive to moderate physiological changes. The authors should temper their statements to “no between-treatment differences in TOC were detected” without inferring photosynthetic performance; or conduct chlorophyll fluorescence essays, to substantiate any claims about photoactivity.
General note and recommendations:
A notable limitation of the work is the absence of critical controls (HTCC-only, phosphate-only at matched P concentration, and ionic iron (e.g. FeSO₄/Fe-EDTA).
I emphasize the importance of publishing careful studies that honestly report the absence of both positive and negative effects. For future work, I recommend colloid–chemical testing of the nanomaterials (e.g., DLS), assessing bioavailability via tissue elemental content, and quantifying chlorophyll activity via chlorophyll fluorescence.
Author Response
Medellín, November 1, 2025.
Editorial Board
Plants
Multidisciplinary Digital Publishing Institute MDPI
Best regard,
The authors of the manuscript entitled “Effects of Functionalized Iron Oxide Magnetic Nanoparticle Suspensions on Seed Morphology and Physiology in Yellow Maize and Chili Pepper” (ID: plants-3927373) would like to thank the reviewers for their valuable comments and suggestions, which have greatly contributed to improving the scientific quality of our work. The following paragraphs describe how each of these comments has been addressed. All corrections have been highlighted in blue in the revised manuscript for ease of identification.
Thank you very much, and please, let us know if any more corrections or clarifications are needed.
Sincerely yours,
The authors.
Reviewer 1
Comments and Suggestions for Authors
The work is devoted to the synthesis and study of phytotoxicity of magnetite-maghemite NPs functionalized with quaternized chitosan, and adsorbed on their surface phospahate. The topic fits within plant physiology, agronomy/agrochemistry, and the emerging field of agricultural nanomaterials.
Biological assays comprised two in vitro phytotoxicity tests. In vitro tests consisted of incubating maize and chili pepper seeds with bare nanocomposite suspension (12.5-100 ppm) or with adsorbed phosphate (2.9-5.2 ppm). They were followed by a greenhouse experiment where the germinated seedlings primed with nanocomposite suspension with adsorbed phosphate were grown in pots for 28 days and their biomass, root length/thickness, stem thickness, seedling height, and longest leaf length were measured. Lastly, total oxidizable organic carbon of the greenhouse-derived seedlings was quantified using the Walkley–Black method.
Across in vitro assays, maize and chilli pepper germination, biomass of the seedlings and root length did not change compared to the control up to 50 ppm of nanocomposite. In the subsequent greenhouse test, plant growth parameters also showed no treatment–control differences. These findings support the overall conclusion that ≤45 ppm Fe and ≤5.2 ppm P were neither phytotoxic nor growth-promoting in these species. The obtained results are useful for early-stage risk assessment and dose selection in seed-priming/coating studies and for safety evaluation frameworks for agricultural nanomaterials.
Major comments:
The authors should explicitly justify the selection of a magnetite–maghemite over alternative iron oxides (e.g., hematite, magnetite-only), outlining the hypothesized advantages.4
Author’s response: magnetite and its high oxidized state of maghemite are preferred in this research due to their higher biocompatibility and magnetic response compared to other iron oxides such as hematite (α-Fe2O3), goethite (α-FeOOH) and akageneite (β-FeOOH) [20-22]. Even though the synthesis method employed in this study was intended to produce a single phase of magnetite, the natural oxidation of magnetite to maghemite is inevitable under atmospheric conditions after the synthesis process. As a result, a mixture of both phases is commonly observed. On the other hand, the magnetic response of magnetite and maghemite is very convenient for applications in nanomedicine such as drug delivery, magnetic hyperthermia and directing these nanomaterials to specific places in alive systems. Hematite, goethite and akageneite have antiferromagnetic ordering, having a very low magnetic response for biotechnology applications. This clarification was introduced in the introductory section of the manuscript.
At a maximum of 5.2 ppm P in the working suspensions (Table 1) and 3 mL per Petri dish, these are microgram-level doses; the authors should either justify that such P input is physiologically meaningful for priming/early growth or treat it as a limitation and consider increasing loading/concentration and quantifying release/uptake.
Author’s response: the phosphorus concentration in the nanocomposites is limited by the adsorption of this element onto their surface. According to the coprecipitation synthesis method employed by us, the maximum adsorbed phosphorus concentration in the original suspension was 40 ppm and the maximum adsorbed phosphorus in the diluted suspension with the maximum iron concentration was 5.2 ppm. This phosphorus concentration is limited by the capacity of the nanocomposites to adsorb phosphate groups onto their surface. Although the maximum phosphorus concentration of the diluted nanocomposites is lower than the 20–40 ppm phosphorus concentrations recommended for fertilization processes [25], this study allows us to determine whether the nanocomposites offer any advantage as macronutrient carriers at lower concentrations compared with conventional fertilizers. This clarification was introduced in Section 2.6 of the manuscript.
The outcomes for maize and chili (Fig 4) should be replot as separate graphs using comparable y-scales within each species; the large between-species magnitude differences makes the current combined plot very hard to read.
Author’s response: thank you very much for this comment, the plots of maize and chili pepper were presented separately.
For each treatment in germination of maize seeds tests, raw replicate points and the mean with 95% confidence intervals should be overlayed (1), and state the CI method should be pointed in the caption. Based on that, the statistical evidence of the character of the dependence of dose–response should be found (is there an extremum still?)
Author’s response: thank you for your suggestion. In performing this exercise, we have observed that, although the p-value of less than 0.05 in the regression model suggests a non-random relationship between the variables evaluated, we cannot affirm that there is a greater or lesser possible effect on the morphological response of the seeds caused by the doses of nanoparticles evaluated, since the overlap of the 95% confidence intervals of the means indicates that there is no statistical evidence to conclude that there is a difference between the treatments. According to this result, the graphs and their interpretation have already been included in the manuscript.
The authors’ conclusions about photosynthetic activity based on total oxidizable organic carbon (TOC) measured by the Walkley–Black method are not justified. Walkley–Black method is a soil wet-oxidation assay not validated for plant tissues as a proxy for photosynthesis, and TOC% is largely insensitive to moderate physiological changes. The authors should temper their statements to “no between-treatment differences in TOC were detected” without inferring photosynthetic performance; or conduct chlorophyll fluorescence essays, to substantiate any claims about photoactivity.
Author’s response: thank you very much for this observation. The conclusions on total oxidizable carbon were guided by the possible effect of different concentrations of nanoparticles on the carbon composition of plant tissues, which could be related to their development and growth. The respective modifications in the interpretation of the results were included in the manuscript.
General note and recommendations:
A notable limitation of the work is the absence of critical controls (HTCC-only, phosphate-only at matched P concentration, and ionic iron (e.g. FeSO₄/Fe-EDTA).
Author’s response: This is true; nevertheless, the scope of this research was limited to evaluating the magnetite–maghemite@quaternized-chitosan@phosphates nanocomposite. The evaluation of treatments with only quaternized chitosan, FeSO₄, or a positive control using commercial fertilizers will be part of future work.
I emphasize the importance of publishing careful studies that honestly report the absence of both positive and negative effects. For future work, I recommend colloid–chemical testing of the nanomaterials (e.g., DLS), assessing bioavailability via tissue elemental content, and quantifying chlorophyll activity via chlorophyll fluorescence.
Author’s response: thank you very much for this comment, in Section 3.1.3 we have included zeta potential and dynamic light scattering measurements of the suspensions of nanocomposites, these measurements show an adequate positive charge on the surface of the nanocomposites which has relationship with a good colloidal stability of the suspension. Chlorophyll analysis is part of future work.
Reviewer 2 Report
Comments and Suggestions for Authors
Please, find here my evaluation report to improve the quality of this research study:
General comments:
The authors have assembled a well-executed characterization of their magnetite-maghemite nanocomposites and structured their experiments thoughtfully. Yet reading through this manuscript, I find myself wanting more from a study that promises to advance our understanding of iron-based nanomaterials in agriculture. The core finding here is reassuring: keeping iron concentrations below 100 ppm avoids phytotoxicity in both maize and chili pepper. That's useful information, certainly, but it addresses only half the equation. We learn what doesn't harm these plants, but not whether the nanocomposites actually help them. Given the considerable effort invested in synthesizing and functionalizing these materials, the absence of beneficial effects deserves deeper examination. Why didn't the nanocomposites deliver on their agricultural promise? The experimental approach suffers from constraints that limit confidence in the conclusions. Sample sizes remain modest throughout, making it difficult to distinguish genuine biological responses from statistical noise. More critically, the study stops at the plant surface, and we never learn whether these nanoparticles enter root tissues, move into shoots, or interact with cellular machinery. Without tracking uptake and translocation, the mechanisms driving these null results remain speculative. This work adds another data point to an increasingly crowded but puzzling literature on iron oxide nanoparticles in agriculture, where contradictory findings abound. The authors acknowledge this complexity and correctly emphasize that outcomes depend heavily on both plant species and nanoparticle formulation. What's missing is a critical discussion of their experimental design choices and how specific limitations might explain the lack of observed benefits. I believe that major revisions are needed. The manuscript requires expanded mechanistic investigations, larger sample sizes for meaningful statistical power, and honest reflection on why this particular formulation failed to enhance plant performance.
Detailed comments:
In this research study, the authors tackle an intriguing question: how do magnetite-maghemite nanocomposites—coated with quaternized chitosan and phosphates—affect seed germination and early growth in maize and chili pepper? The authors ground their work thoughtfully within the expanding field of agricultural nanotechnology, acknowledging something that often gets glossed over: the existing literature on iron oxide nanoparticles is riddled with contradictions. Some studies report benefits, others warn of toxicity, and many fall somewhere in between. What makes this investigation particularly worthwhile is its focus on the knowledge gap that chitosan-coated iron oxide nanoparticles remain understudied in maize and chili pepper. The stated objectives are straightforward: evaluate how these nanocomposites influence morpho-anatomical features and physiological responses in seedlings. The aims are well defined: to assess morpho-anatomical and physiological parameters after nanocomposite application. But the introduction could be improved by stating specific hypotheses beyond the exploratory goal.
The authors build their case methodically, moving from nanocomposite synthesis and characterization through phytotoxicity assays and ultimately to greenhouse trials. This progression makes sense: starting with dose-response screening, narrowing to targeted in vitro work, then scaling up to more realistic growing conditions. One could argue this reflects thoughtful experimental design. That said, a few structural choices interrupt the narrative's natural rhythm. Burying the bulk of characterization data in supplementary materials creates an awkward gap in the story. Readers encounter biological experiments without first seeing the essential physical and chemical properties of what's being tested. Even a condensed presentation of particle size distributions, compositional analysis, and functionalization verification would anchor the later discussions more effectively. These aren't optional details, they're the foundation for interpreting any biological response. Furthermore, the analysis of organic carbon measurements (Section 3.3) seems somewhat detached from the primary discussion of phytotoxicity and could benefit from better integration.
The authors describe their nanocomposite synthesis comprehensively, drawing on XRD, FTIR, Mössbauer spectroscopy, TEM, and magnetization measurements. This multi-technique approach provides convincing material characterization, while the seed preparation protocols demonstrate appropriate attention to sterilization. The statistical methodology appears sound, but the sample sizes strike me as modest. In vitro assays relied on just three replicates per treatment, while greenhouse trials used three maize plants and four chili peppers. Given how variable biological systems can be, these numbers may not capture subtle treatment effects or provide sufficient statistical power. The control group setup also raises questions. The initial in vitro assay used tap water as a control, whereas later assays seem to switch to deionized water. This inconsistency could muddy the interpretation of results and it's unclear whether this was intentional or simply an oversight that needs clarification.
More fundamentally, the study leaves some methodological gaps unexplored. Take the in vitro germination assays: while the authors mention a "dark room," we don't know the exact lighting conditions or duration. There's also no positive control (conventional iron or phosphorus fertilizers, for instance) which would help benchmark whether the nanocomposite outperforms standard treatments. The greenhouse experiments lack similar details about irrigation schedules, substrate composition, and growing conditions beyond temperature. Perhaps most surprisingly, despite measuring various morphological parameters, the study doesn't track whether nanoparticles actually enter the plants or where they accumulate. The phosphorus adsorption data deserve mention too. The maximum concentration reached only 5.2 ppm, which seems quite low for practical agricultural applications. The authors acknowledge this limitation, though they don't really address whether such concentrations would meaningfully benefit crops in field conditions. Finally, the 28-day greenhouse evaluation window may be too brief. Many physiological responses (particularly stress responses or nutrient-related changes) can take time to manifest, and a longer observation period might reveal effects that weren't apparent in this timeframe.
The data presentation shows clear strengths, particularly in the regression analysis (Figure 1), which nicely captures the dose-dependent relationship between iron concentration and both biomass and root length. Statistical analyses appear sound, and the figures themselves are well-labeled. That said, a few aspects of the data handling left me wondering about their implications. The germination percentage story puzzles me somewhat. Maize germination dropped noticeably between the first assay (over 93%) and the second (70-93%), chalked up rather vaguely to "seed condition." This isn't trivial variation, it's substantial enough to make me question what's happening with seed quality control or storage protocols. Without understanding this shift better, it becomes tricky to separate genuine treatment effects from these background inconsistencies. The authors conclude, reasonably, that nanocomposites below 100 ppm iron don't trigger phytotoxic responses. But I'm left wondering if they're being too cautious about the lack of beneficial effects. The study essentially reports null results (treatments didn't differ meaningfully from controls) yet several questions remain unaddressed. Did the experiment have adequate statistical power to detect biologically meaningful differences if they existed? Were the chosen concentrations and delivery methods truly optimal for nutrient uptake? Might alternative approaches, like foliar application or soil amendments, tell a different story? The organic carbon data (Table 2) shows modest variation between treatments (30.5-37.5%), but without statistical testing, it's hard to know whether these differences matter. The qualitative leap to suggesting that similar organic carbon levels indicate "good photosynthetic capacity" seems to need more rigorous backing.
Finally, what's puzzling me is the absence of any real mechanistic exploration. When nanocomposites show neither benefit nor harm (essentially doing nothing) that's a result worth interrogating. Were the particles simply not bioavailable? Did the plants fail to take them up? Could the chitosan coating have acted as a barrier rather than a facilitator? The phosphorus content itself might have been too modest to make a difference. These possibilities hang in the air, unexamined.
Author Response
Medellín, November 1, 2025.
Editorial Board
Plants
Multidisciplinary Digital Publishing Institute MDPI
Best regard,
The authors of the manuscript entitled “Effects of Functionalized Iron Oxide Magnetic Nanoparticle Suspensions on Seed Morphology and Physiology in Yellow Maize and Chili Pepper” (ID: plants-3927373) would like to thank the reviewers for their valuable comments and suggestions, which have greatly contributed to improving the scientific quality of our work. The following paragraphs describe how each of these comments has been addressed. All corrections have been highlighted in blue in the revised manuscript for ease of identification.
Thank you very much, and please, let us know if any more corrections or clarifications are needed.
Sincerely yours,
The authors.
Reviewer 2
General comments:
The authors have assembled a well-executed characterization of their magnetite-maghemite nanocomposites and structured their experiments thoughtfully. Yet reading through this manuscript, I find myself wanting more from a study that promises to advance our understanding of iron-based nanomaterials in agriculture. The core finding here is reassuring: keeping iron concentrations below 100 ppm avoids phytotoxicity in both maize and chili pepper. That's useful information, certainly, but it addresses only half the equation. We learn what doesn't harm these plants, but not whether the nanocomposites actually help them. Given the considerable effort invested in synthesizing and functionalizing these materials, the absence of beneficial effects deserves deeper examination. Why didn't the nanocomposites deliver on their agricultural promise? The experimental approach suffers from constraints that limit confidence in the conclusions. Sample sizes remain modest throughout, making it difficult to distinguish genuine biological responses from statistical noise. More critically, the study stops at the plant surface, and we never learn whether these nanoparticles enter root tissues, move into shoots, or interact with cellular machinery. Without tracking uptake and translocation, the mechanisms driving these null results remain speculative. This work adds another data point to an increasingly crowded but puzzling literature on iron oxide nanoparticles in agriculture, where contradictory findings abound. The authors acknowledge this complexity and correctly emphasize that outcomes depend heavily on both plant species and nanoparticle formulation. What's missing is a critical discussion of their experimental design choices and how specific limitations might explain the lack of observed benefits. I believe that major revisions are needed. The manuscript requires expanded mechanistic investigations, larger sample sizes for meaningful statistical power, and honest reflection on why this particular formulation failed to enhance plant performance.
Detailed comments:
In this research study, the authors tackle an intriguing question: how do magnetite-maghemite nanocomposites—coated with quaternized chitosan and phosphates—affect seed germination and early growth in maize and chili pepper? The authors ground their work thoughtfully within the expanding field of agricultural nanotechnology, acknowledging something that often gets glossed over: the existing literature on iron oxide nanoparticles is riddled with contradictions. Some studies report benefits, others warn of toxicity, and many fall somewhere in between. What makes this investigation particularly worthwhile is its focus on the knowledge gap that chitosan-coated iron oxide nanoparticles remain understudied in maize and chili pepper. The stated objectives are straightforward: evaluate how these nanocomposites influence morpho-anatomical features and physiological responses in seedlings. The aims are well defined: to assess morpho-anatomical and physiological parameters after nanocomposite application. But the introduction could be improved by stating specific hypotheses beyond the exploratory goal.
Author’s response: Thank you very much for this comment, the introductory section concludes by stating that the goal and scope of the research is to address two questions relevant to the use of nanoparticles in plants, namely: 1. Are magnetite–maghemite nanoparticles functionalized with quaternized chitosan biocompatible with yellow hybrid maize and chili pepper seeds within a given iron concentration range? 2. Does the functionalization of magnetite–maghemite nanoparticles with quaternized chitosan and phosphates enhance the morphological and physiological parameters of seeds and their subsequent seedlings compared with the water control? The authors consider that these two questions guide and define the scope of the research beyond the exploratory purpose.
The authors build their case methodically, moving from nanocomposite synthesis and characterization through phytotoxicity assays and ultimately to greenhouse trials. This progression makes sense: starting with dose-response screening, narrowing to targeted in vitro work, then scaling up to more realistic growing conditions. One could argue this reflects thoughtful experimental design. That said, a few structural choices interrupt the narrative's natural rhythm. Burying the bulk of characterization data in supplementary materials creates an awkward gap in the story. Readers encounter biological experiments without first seeing the essential physical and chemical properties of what's being tested. Even a condensed presentation of particle size distributions, compositional analysis, and functionalization verification would anchor the later discussions more effectively. These aren't optional details, they're the foundation for interpreting any biological response.
Author’s response: thank you very much for this observation, of course, the synthesis and characterization of the nanocomposite required a significant effort in this research and represents a relevant problem in materials science, which was not sufficiently emphasized in the initial version of the manuscript. In response to this valuable recommendation, we have included in Section 3.1 a detailed description of the structural, chemical, morphological, and magnetic measurements performed on the nanocomposite to determine its main properties prior to its application on maize and chili pepper seeds.
Furthermore, the analysis of organic carbon measurements (Section 3.3) seems somewhat detached from the primary discussion of phytotoxicity and could benefit from better integration.
Author’s response: thank you very much for this observation; the Total Oxidizable Carbon discussion was addressed to know if the treatments with the composite have some indirect influence on the overall physiological development of the plants evaluated. The respective modifications in the interpretation of the results and the relationship with previous test results have already been included in the manuscript.
The authors describe their nanocomposite synthesis comprehensively, drawing on XRD, FTIR, Mössbauer spectroscopy, TEM, and magnetization measurements. This multi-technique approach provides convincing material characterization, while the seed preparation protocols demonstrate appropriate attention to sterilization. The statistical methodology appears sound, but the sample sizes strike me as modest. In vitro assays relied on just three replicates per treatment, while greenhouse trials used three maize plants and four chili peppers. Given how variable biological systems can be, these numbers may not capture subtle treatment effects or provide sufficient statistical power.
Author’s response: this is true, the statistical design of the experiment was based on previous studies that reported the minimum number of individuals and replicates required per seed and seedling analyzed. However, it is possible that using 10–20 individuals per treatment with three replicates was insufficient to detect significant differences between the treated plants and the control. This limitation was emphasized in the final conclusions of the manuscript, and it may serve as a useful reference to guide the design of similar experiments in the future.
The control group setup also raises questions. The initial in vitro assay used tap water as a control, whereas later assays seem to switch to deionized water. This inconsistency could muddy the interpretation of results and it's unclear whether this was intentional or simply an oversight that needs clarification.
Author’s response: thank you very much for this comment. This was an error on our part. As mentioned in the methodology section, deionized water was used as a control for both the first and second in vitro assay. We have already made the necessary corrections.
More fundamentally, the study leaves some methodological gaps unexplored. Take the in vitro germination assays: while the authors mention a "dark room," we don't know the exact lighting conditions or duration. There's also no positive control (conventional iron or phosphorus fertilizers, for instance) which would help benchmark whether the nanocomposite outperforms standard treatments.
Author’s response: thank you very much for this observation, by dark room we mean a room with the lowest possible level of light, with all windows closed to minimize seed exposure to illumination. This term does not refer to any special or controlled lighting conditions. This clarification has been included in the manuscript.
Regarding the use of a positive control, it would have been desirable to compare the development of seeds and seedlings with such a treatment. Nevertheless, the scope of this research was primarily focused on assessing the biocompatibility of iron oxide nanocomposites and the possible effect in morphological and physiological parameters of maize and chili pepper seeds when phosphate groups are adsorbed on their surface.
The greenhouse experiments lack similar details about irrigation schedules, substrate composition, and growing conditions beyond temperature.
Author’s response: Thank you for this comment. In the case of the commercial substrate, it was specified that it is black soil (Mixture of organic soil, decomposed plant matter, and essential minerals; obtained from Tierragro), which allows for greater water retention, is rich in essential nutrients, and is loose. As for irrigation conditions, watering was done once a day with a specific volume of tap water for each seedling, and at the end of the day, the soil was checked to ensure it was not dry. These conditions have already been included in the manuscript.
Perhaps most surprisingly, despite measuring various morphological parameters, the study doesn't track whether nanoparticles actually enter the plants or where they accumulate.
Author’s response: this is entirely true and constitutes a future stage of our research. The translocation of nanoparticles from the site of application to various plant tissues is a crucial and determining factor in assessing their true physiological effects.
The phosphorus adsorption data deserve mention too. The maximum concentration reached only 5.2 ppm, which seems quite low for practical agricultural applications. The authors acknowledge this limitation, though they don't really address whether such concentrations would meaningfully benefit crops in field conditions.
Author’s response: the phosphorus concentration in the nanocomposites is limited by the adsorption of this element onto their surface. According to the coprecipitation synthesis method employed by us, the maximum adsorbed phosphorus concentration in the original suspension of nanocomposite was 40 ppm and the maximum adsorbed phosphorus in the diluted suspension with the maximum iron concentration was 5.2 ppm. Certainly, it is lower than the range of phosphorus concentration levels (20-40 ppm) recommended for fertilization processes [25]. Nevertheless, one of the aims of this study was to determine whether the high surface-to-volume ratio of the nanoparticles within the nanocomposite confers any advantage as a macronutrient carrier at phosphorus concentrations below the reference levels. This effect could not be verified in the morphological and physiological parameters measured, possibly due to the low statistical power, the short monitoring period after seed germination, or both factors. This clarification has been incorporated into Section 2.6 of the manuscript.
Finally, the 28-day greenhouse evaluation window may be too brief. Many physiological responses (particularly stress responses or nutrient-related changes) can take time to manifest, and a longer observation period might reveal effects that weren't apparent in this timeframe.
Author’s response: This is true, and long-term effects may appear as a result of the treatments. However, the main objective of this study was to evaluate the possible phytotoxic effect of iron oxide nanoparticles functionalized with chitosan and phosphates adsorbed on their surface on maize and chili pepper seeds, and subsequently verify that these seeds, which were in contact with the treatments, had adequate initial vegetative development, focusing mainly on their germination, initial root development, and stem and first leaf formation.
The data presentation shows clear strengths, particularly in the regression analysis (Figure 1), which nicely captures the dose-dependent relationship between iron concentration and both biomass and root length. Statistical analyses appear sound, and the figures themselves are well-labeled. That said, a few aspects of the data handling left me wondering about their implications. The germination percentage story puzzles me somewhat. Maize germination dropped noticeably between the first assay (over 93%) and the second (70-93%), chalked up rather vaguely to "seed condition." This isn't trivial variation, it's substantial enough to make me question what's happening with seed quality control or storage protocols. Without understanding this shift better, it becomes tricky to separate genuine treatment effects from these background inconsistencies.
Author’s response: thank you very much for this observation; In our study, the seeds we received for the trials were stored in an airtight, clean, dry container, which was kept in a cool, dry, dark, pest-free place, a special space for seed storage, and before using these seeds, their quality was checked to ensure they had appropriate viability. Maize seeds were provided by an entity that supplied us with small batches whenever we needed them. In this regard, one possible reason for the difference between the two trials was that the two germination assays were conducted using different seed batches, supplied sequentially by the provider as our experimental work progressed. The first batch used, which was used for the first in vitro test, was also used for other tests not reported in this paper. Since we were running out of seeds for the following tests, the entity provided us with a new batch of maize seeds. Although both batches met the standard viability threshold and were handled under the same storage and experimental conditions, variation between seed lots may have caused physiological heterogeneity in the germination tests. Nevertheless, we recognize that variation between seed lots can introduce background variability, and it will be appropriate for future work to include the lot as a blocking factor in the experimental design to obtain more adequate experimental results.
The authors conclude, reasonably, that nanocomposites below 100 ppm iron don't trigger phytotoxic responses. But I'm left wondering if they're being too cautious about the lack of beneficial effects.
Author’s response: the conclusions highlight the positive effects of the nanocomposites in the root length and dry biomass of yellow hybrid maize for iron concentrations lower than 50 ppm. Nevertheless, as well expressed by the reviewer, more measurements must be performed to point out positive effects in other indicators, among them chlorophyll concentration, nutrient concentration in tissues, among others.
The study essentially reports null results (treatments didn't differ meaningfully from controls) yet several questions remain unaddressed. Did the experiment have adequate statistical power to detect biologically meaningful differences if they existed? Were the chosen concentrations and delivery methods truly optimal for nutrient uptake? Might alternative approaches, like foliar application or soil amendments, tell a different story?
Author’s response: thank you for your comments. Throughout the manuscript and based on your suggestions in this evaluation, we have found that in order to achieve more accurate and clear results in the different trials carried out, it is necessary to use a much larger number of replicates than those used in this study in order to have sufficient statistical power to support the results. In addition to this, it would be necessary to have a longer evaluation time in the case of the greenhouse trial to allow the plant to show a biological response to the treatments to which it was subjected. These considerations and new experimental trials that evaluate the physiological response of plants to treatments will be considered for future work.
The organic carbon data (Table 2) shows modest variation between treatments (30.5-37.5%), but without statistical testing, it's hard to know whether these differences matter. The qualitative leap to suggesting that similar organic carbon levels indicate "good photosynthetic capacity" seems to need more rigorous backing.
Author’s response: thank you very much for this observation, the conclusions on the Total Oxidizable Carbon were oriented to the similar development of plants of different treatments concerned with their overall physiological processes.
Finally, what's puzzling me is the absence of any real mechanistic exploration. When nanocomposites show neither benefit nor harm (essentially doing nothing) that's a result worth interrogating. Were the particles simply not bioavailable? Did the plants fail to take them up? Could the chitosan coating have acted as a barrier rather than a facilitator? The phosphorus content itself might have been too modest to make a difference. These possibilities hang in the air, unexamined.
Thank you very much for this valuable comment. The main result of this research is that magnetite–maghemite nanocomposites functionalized with quaternized chitosan, either alone or combined with phosphate groups, did not produce adverse effects on the seeds of the two evaluated plant species, even at iron concentrations higher than those used in other studies where bare nanoparticles caused toxic effects. This finding indicates that functionalization of magnetite–maghemite nanoparticles with this biocompatible polymer improves their compatibility with plant tissues and enhances both root length and fresh biomass of germinated seeds. Moreover, the magnetic nature of magnetite–maghemite nanoparticles offers future possibilities for exploring the delivery of micro- and macronutrients adsorbed onto their surface, potentially assisted by the application of external magnetic fields.

Reviewer 3 Report
Comments and Suggestions for Authors
The current research aimed to evaluate the effects of Iron (Fe) nanoparticles on morpho-anatomical and Physiological traits of maize and chili pepper. The authors attempted to demonstrate and conclude that aqueous suspensions of magnetite–maghemite nanocomposites with Fe concentrations below 100 ppm do not produce phytotoxic effects on seed germination or morpho-anatomical and physiological traits.
Overall, the Research was conducted well; the results supported the hypothesis.
However, there are still some opportunities to improve the manuscript before final publication:
1) In the abstract/Conclusion: It's claimed/summarized that the Fe concentrations below 100 ppm do not produce phytotoxic effects. Nonetheless, there was no gradual treatment to evaluate the dosage effect between 75 ppt and 100 ppm, and above 100 ppm. It may be suitable to narrow down the range to get an optimum concentration.
2) In the introduction section, the dosage and concentration effect may need to be elaborated further.
3) in Methods:
- (specifically, statistical analysis)The effect of phosphorus (P) along with Fe seems to be ignored. The interaction of Fe and P may also need to be estimated in ANOVA.
- for HClO4: it's better to add its CAS number/supplier due to its hazardous nature.
- L98: “with some modifications”: Its better to specify the modifications and the reasons to modifiy.
- L102: “was precipitated of the reaction solution”: it may be “was precipitated from the reaction solution”
- L104: During the Magnetic Nanocomposite preparation, the pH control, stirring conditions (rpm, magnetics or mechanical stirrer), and the nitrogen purity level may need to be specified.
- f) L112: “suspension was obtained”. pH of the final suspension and its stability time (before sedimentation) may need to be reported
4) In the Results and Discussion section
4.1: L231: Section 3.2.1 is too long and mixes the introduction and methodology too. It would be better to divide it into segments and make the theme brief to understand.
4.2: Overall, this section should be reviewed for clarity of statements, specifically against statistical arguments (for example: L274 “parabolic fitting model”, can be “parabolic regression model”), structure of paragraphs (One paragraph for one theme), interpretation of results along with literature and its background information.
4.4: Try to cite the tables and figures immediately wherever its discussed.
5: Finally, minor linguistic corrections may require and need to be consistent throughout the manuscript. Some of those are as follows:
L98: “Rihua et al [20]” may need to be cited properly
L156: Change from “high quantity” to “high content” or “high concentration”
L159: Scientific names should be italic
L219: better to change from “highly oxidate state” to “highly oxidized state”
L225-226: Change “measurements showed the behavior of a soft magnetic” to “measurements indicate soft magnetic.."
L221: Change “quasi-spherical shape” to “nearly spherical morphology”
L228: Change “low size of the particles” to “small particle size”
L279: change “were studied” to “were evaluated”
L313: can change from “percentage achieved by the maize with respect to the first in vitro assay” to the “germination percentage observed in maize compared to the first assay”
Author Response
Medellín, November 1, 2025.
Editorial Board
Plants
Multidisciplinary Digital Publishing Institute MDPI
Best regard,
The authors of the manuscript entitled “Effects of Functionalized Iron Oxide Magnetic Nanoparticle Suspensions on Seed Morphology and Physiology in Yellow Maize and Chili Pepper” (ID: plants-3927373) would like to thank the reviewers for their valuable comments and suggestions, which have greatly contributed to improving the scientific quality of our work. The following paragraphs describe how each of these comments has been addressed. All corrections have been highlighted in blue in the revised manuscript for ease of identification.
Thank you very much, and please, let us know if any more corrections or clarifications are needed.
Sincerely yours,
The authors.
Reviewer 3
Comments and Suggestions for Authors
The current research aimed to evaluate the effects of Iron (Fe) nanoparticles on morpho-anatomical and Physiological traits of maize and chili pepper. The authors attempted to demonstrate and conclude that aqueous suspensions of magnetite–maghemite nanocomposites with Fe concentrations below 100 ppm do not produce phytotoxic effects on seed germination or morpho-anatomical and physiological traits.
Overall, the Research was conducted well; the results supported the hypothesis.
However, there are still some opportunities to improve the manuscript before final publication:
1) In the abstract/Conclusion: It's claimed/summarized that the Fe concentrations below 100 ppm do not produce phytotoxic effects. Nonetheless, there was no gradual treatment to evaluate the dosage effect between 75 ppm and 100 ppm, and above 100 ppm. It may be suitable to narrow down the range to get an optimum concentration.
Author’s response: yes, it would have been desirable to evaluate the effect of treatments with intermediate iron concentrations between 75 ppm and 100 ppm, however, the experiment was designed to variate in steps of 25 ppm after 50 ppm. Unfortunately, these measurements would have to be carried out as future work.
2) In the introduction section, the dosage and concentration effect may need to be elaborated further.
Author’s response: Thank you very much for your comment. The introduction mentions some references that show the effect of different concentrations and doses of nanoparticles on various plants.
3) in Methods:
- (specifically, statistical analysis) The effect of phosphorus (P) along with Fe seems to be ignored. The interaction of Fe and P may also need to be estimated in ANOVA. Author’s response: Thank you very much for your suggestion. This will be taken into account in future work to carry out experiments with the appropriate controls and parameters to include these variables in the evaluations.
- for HClO4: it's better to add its CAS number/supplier due to its hazardous nature. Author’s response: the CAS number of the HClO4 was included in section 2.1.
- L98: “with some modifications”: Its better to specify the modifications and the reasons to modifiy.
Author’s response: the only modification consisted of adding 8.0 mL of GTMAC in two portions at 0.5 h intervals to the chitosan–HClO₄ solution, instead of in three portions. The procedure described in Section 2.2 already includes all the steps required to perform this process; therefore, we have removed the phrase “with some modifications” to prevent any confusion for the reader.
- L102: “was precipitated of the reaction solution”: it may be “was precipitated from the reaction solution”
Author’s response: thank you very much, the change was introduced in the manuscript.
- L104: During the Magnetic Nanocomposite preparation, the pH control, stirring conditions (rpm, magnetics or mechanical stirrer), and the nitrogen purity level may need to be specified.
Author’s response: details of the pH control, purity level of N2(g) and magnetic stirring were included in section 2.3.
- f) L112: “suspension was obtained”. pH of the final suspension and its stability time (before sedimentation) may need to be reported.
Author’s response: the final pH of the suspension after washing was reported, as well as the evidence of colloidal stability of the final product supported by Zeta potential measurements.
4) In the Results and Discussion section
4.1: L231: Section 3.2.1 is too long and mixes the introduction and methodology too. It would be better to divide it into segments and make the theme brief to understand.
Author’s response: introductory and redundant comments were removed from Section 3.2.1, and the section was focused on the results and discussion.
4.2: Overall, this section should be reviewed for clarity of statements, specifically against statistical arguments (for example: L274 “parabolic fitting model”, can be “parabolic regression model”), structure of paragraphs (One paragraph for one theme), interpretation of results along with literature and its background information.
Author’s response: The suggested modifications have already been included in the manuscript
4.4: Try to cite the tables and figures immediately wherever its discussed.
Author’s response: tables and figures were placed as close as possible to the paragraphs in which they are cited.
5: Finally, minor linguistic corrections may require and need to be consistent throughout the manuscript. Some of those are as follows:
L98: “Rihua et al [20]” may need to be cited properly: the correction of Rihua to Ruihua was made.
L156: Change from “high quantity” to “high content” or “high concentration”: the suggested correction was made.
L159: Scientific names should be italic: capsicum annuum L. was written in italics.
L219: better to change from “highly oxidate state” to “highly oxidized state”: the correction of oxidate to oxidized was made.
L225-226: Change “measurements showed the behavior of a soft magnetic” to “measurements indicate soft magnetic.": the correction was made.
L221: Change “quasi-spherical shape” to “nearly spherical morphology”: the correction was made.
L228: Change “low size of the particles” to “small particle size”: the correction was made.
L279: change “were studied” to “were evaluated”: the correction was made.
L313: can change from “percentage achieved by the maize with respect to the first in vitro assay” to the “germination percentage observed in maize compared to the first assay”: the correction was made.

Round 2
Reviewer 1 Report
Comments and Suggestions for Authors
All major comments have been thoroughly and appropriately addressed—figures were replotted, statistical treatment was clarified, several interpretations were tempered, and colloid-chemical characterization was added. I recommend the manuscript to be accepted for publication without further changes.
Reviewer 3 Report
Comments and Suggestions for Authors
Authors have made significant changes in revised version of manuscript.